# Cost-Effectiveness Analysis of Axicabtagene Ciloleucel vs. Standard of Care in Second-Line Treatment for Relapsed/Refractory Large B-Cell Lymphoma in Spain

**DOI:** 10.3390/cancers16132301

**Published:** 2024-06-22

**Authors:** Alejandro Martín García-Sancho, María Presa, Carlos Pardo, Victoria Martín-Escudero, Itziar Oyagüez, Valentín Ortiz-Maldonado

**Affiliations:** 1Haematology Department, Instituto de Investigación Biomédica de Salamanca (IBSAL), CIBERONC, Hospital Universitario de Salamanca, University of Salamanca, 37007 Salamanca, Spain; amartingar@usal.es; 2Health Economics Department, Pharmacoeconomics & Outcomes Research Iberia (PORIB), 28224 Madrid, Spain; ioyaguez@porib.com; 3Market Access, Reimbursement & Health Economics and Outcomes Research Department, Gilead Sciences, 28033 Madrid, Spain; carlos.pardo@gilead.com (C.P.); victoria.martin-escudero@gilead.com (V.M.-E.); 4Haematology Department, Hospital Clínic de Barcelona, 08036 Barcelona, Spain; vortiz@clinic.cat

**Keywords:** axicabtagene ciloleucel, large B-cell lymphoma, diffuse large B-cell lymphoma, cost-effectiveness analysis, second-line treatment

## Abstract

**Simple Summary:**

At least 40% of treated patients with large B-cell lymphoma do not respond to first-line treatment or experience disease recurrence, and less than 50% of patients respond to second-line salvage immunochemotherapy and can proceed to the historical standard of care of autologous stem-cell transplantation (ASCT). This study aimed to assess the cost-effectiveness of axicabtagene ciloleucel compared to salvage immunochemotherapy followed by high-dose chemotherapy (HDT) and ASCT for the treatment of large B-cell lymphoma in second-line patients in Spain. Compared to patients treated with HDT+ASCT, patients treated with axicabtagene ciloleucel experienced improvements in health outcomes in terms of life years gained (LYG) (+1.72) and quality-adjusted life years (QALYs) (+1.81). The incremental cost–utility ratio of axicabtagene ciloleucel versus HDT+ASCT was 47,309 EUR/QALY. Axicabtagene ciloleucel could be a cost-effective option that addresses an unmet clinical need for the treatment of relapsed/refractory large B-cell lymphoma after first-line treatment.

**Abstract:**

Purpose: To estimate the cost-effectiveness of axi-cel vs. salvage immunochemotherapy followed by high-dose chemotherapy and autologous stem-cell transplantation (HDT+ASCT) for responders to second-line treatment for relapsed/refractory (R/R) large B-cell lymphoma (LBCL). Methods: A partitioned survival mixture-cure model comprising three health states was used to estimate the costs, life years gained (LYG), and quality-adjusted life years (QALYs) accumulated over a lifetime horizon. Overall survival, event-free survival, and time to the next treatment with axi-cel and HDT+ASCT were derived from the ZUMA-7 study. The total costs (EUR, 2022) included drug acquisition and administration, ASCT, subsequent treatment, disease and adverse event management, and palliative care. The unitary costs were derived from local databases and the literature. A 3% discount rate was applied to the costs and outcomes. Results: Compared with HDT+ASCT, axi-cel provided higher LYG per patient (10.00 vs. 8.28 LYG/patient) and greater QALYs gained per patient (7.85 vs. 6.04 QALY/patient). The lifetime total costs were 343,581 EUR/patient with axi-cel vs. 257,994 EUR/patient with IQT+ASCT. The incremental cost-effectiveness ratio of axi-cel vs. HDT+ASCT was 49,627 EUR/LYG, and the incremental cost-utility ratio was 47,309 EUR/QALY. Sensitivity analyses confirmed the robustness of the model. Conclusion: Axi-cel is a potentially cost-effective alternative to HDT+ASCT for the treatment of R/R DLBCL in Spain.

## 1. Introduction

Non-Hodgkin lymphoma (NHL) is the most prevalent haematological malignancy worldwide [1]. In Spain, NHL is the sixth most common cancer, with an estimated 9943 new cases in 2023 (3.6% of all new cancer cases) [2].

Large B-cell lymphoma (LBCL) represents the most common lymphoid malignancy (30% of all NHLs) [3] and comprises a heterogeneous group of aggressive B-cell NHLs, including high-grade B-cell lymphoma (HGBL) and diffuse large B-cell lymphoma (DLBCL) [4].

Despite the significant advances in the therapeutic management of B-cell lymphomas, approximately 30–40% of patients with DLBCL do not respond or relapse after standard first-line treatment with R-CHOP (rituximab in combination with cyclophosphamide, doxorubicin, vincristine, and prednisone) [5].

Until recently, the standard of care (SoC) for second-line treatment with curative intent for patients with relapsed or refractory (R/R) LBCL was salvage immunochemotherapy followed by high-dose chemotherapy and autologous stem-cell transplantation (ASCT) in chemosensitive patients [6]. Unfortunately, only 50% of patients respond to second-line salvage immunochemotherapy and can undergo ASCT, with a 3-year event-free survival (EFS) rate of 31% [6].

Anti-CD19 chimeric antigen receptor (CAR) T-cell therapies have revolutionized the treatment of patients with R/R B-cell lymphomas after having demonstrated promising results [7]. Axicabtagene ciloleucel (axi-cel) was approved by the European Medicines Agency (EMA) in June 2018 “for the treatment of adult patients with R/R DLBCL and primary mediastinal large B-cell lymphoma after two or more lines of systemic therapy and adult patients with R/R follicular lymphoma after three or more lines of systemic therapy” [8]. In October 2022, it was approved “for the treatment of adult patients with DLBCL and HGBL who relapse within 12 months after the completion of, or who are refractory to, first-line immunochemotherapy”, based on the results of the ZUMA-7 clinical trial [8,9,10].

ZUMA-7 was a randomized, phase 3 clinical trial comparing axi-cel with the SoC (salvage immunochemotherapy followed by high-dose chemotherapy and ASCT) as a second-line treatment in patients with R/R LBCL [11]. After a median follow-up of 24.9 months (data cut-off 18 March 2021), axi-cel achieved a median EFS of 8.3 months and a response rate of 83% (2.0 months and 50%, respectively, in patients treated with the SoC), with an estimated overall survival (OS) of 61% at 2 years (52% in the SoC) [11].

Given the significant improvements observed with axi-cel, and because economic evaluations have become an important tool for decision-makers, this study aimed to assess the cost-effectiveness of axi-cel compared to the SoC as a second-line treatment for R/R LBCL in Spain.

## 2. Materials and Methods

### 2.1. Model Structure

A cost-effectiveness model developed in Microsoft Excel^®^ (version 2405) and previously published [11] was adapted to estimate the total costs, the life years gained (LYG) and the quality-adjusted life years (QALYs) accrued over a lifetime horizon for a hypothetical cohort of adult patients with LBCL who were refractory or had relapsed after their first-line chemotherapy regimen. Although there was no specific protocol for this study, more details about the model’s development were previously published [12].

According to previously published cost-effectiveness analyses [12,13] and National Institute for Health and Care Excellence (NICE) appraisals [14,15], a partitioned survival mixture-cure model (PS-MCM) with three mutually exclusive health states (event-free, post-event, and death) was considered (Figure 1).

The PS-MCM was the most accurate approach for predicting OS over the long term [16] in a cohort composed of a proportion of patients who were “statistically cured”, with an OS similar to that of an age- and sex-matched general population, and a proportion of patients “not cured”, who were at increased risk of mortality related to R/R LBCL.

All patients were initially in an event-free health state, having had received axi-cel or the SoC, and they remained in that state or progressed, and subsequently received a third-line treatment or died.

The analysis was carried out using a one-month cycle length over a lifetime horizon, which was set as 50 years, to reflect the overall life of the patients. A half-cycle correction was applied [17].

The cost-effectiveness analysis considered the perspective of the Spanish National Health System (NHS) and applied an annual discount rate of 3% to both the costs and the health outcomes, following the most recent recommendations for conducting economic evaluations in Spain [18]. 

A consensus meeting with two haematologists was carried out with the purpose of validating the parameters, the assumptions, and the health care resource use employed in the model. A consensus was reached after discussion, leading to a collective decision.

### 2.2. Population

The R/R LBCL patients who were transplanted intentionally and who relapsed within 12 months of first-line therapy were considered in the analysis, in line with the population included in the ZUMA-7 clinical trial [11].

Characteristics, including the mean patient age (57.2 years) and the proportion of females (39%), were defined based on the patients treated in the ZUMA-7 clinical trial and were used to model long-term survival for patients who achieved a long-term response. Based on the recommendations for the Spanish population, a mean body surface area of 1.70 m^2^ and a mean weight of 70 kg were assumed [19].

### 2.3. Treatment Strategies

The cost-effectiveness model compared the use of axi-cel to salvage immunochemotherapy followed by high-dose chemotherapy and ASCT in chemosensitive patients, as the second-line treatment of R/R LBCL. In line with the clinical practice in Spain, salvage immunochemotherapy comprised R-DHAP (rituximab, dexamethasone, high-dose cytarabine, cisplatin), R-ESHAP (rituximab, etoposide, methylprednisolone, cisplatin, cytarabine), R-GDP (rituximab, gemcitabine, dexamethasone, cisplatin), and R-ICE (rituximab, ifosfamide, carboplatin, etoposide). 

Subsequent treatments included chemotherapy (R-ESHAP, R-GDP, polatuzumab vedotin plus bendamustine and rituximab, and a combination of methylprednisolone and cyclophosphamide), CAR T-cell therapy, palliative treatment, and even clinical trials.

### 2.4. Clinical Data

Transitions into each of the three health states were defined based on the EFS and OS Kaplan–Meier curves (data cut-off 18 March 2021; median follow-up of 24.9 months) from the ZUMA-7 clinical trial [11]. The initiation of subsequent treatment in the post-event health state was defined by the time to next treatment (TTNT) curve.

Based on the evidence about long-term LBCL survivors, the model assumed that patients who remained in an event-free state for at least five years had an OS comparable to that of the general population [20].

The axi-cel and SoC OS curves were capped based on the expected age- and sex-matched survival of the general Spanish population, employing pre-2020 mortality rates to avoid excess mortality associated with the COVID-19 pandemic [21].

### 2.5. Statistical Methods

To extrapolate health outcomes beyond the clinical trial follow-up period and distribute patients among the health states over a lifetime horizon, seven survival functions (exponential, Gompertz, log-normal, log-logistic, gamma, generalized gamma, and Weibull) were fitted to the EFS, OS, and TTNT data. 

Based on the Akaike and Bayesian information criteria, which provide an indication of the statistical goodness of fit and the best clinical plausibility of long-term extrapolation, the log-logistic, generalized gamma, and log-logistic hazard for the uncured fraction were chosen for the axi-cel arm’s EFS, OS, and TTNT, respectively. 

For the SoC, the MCMs obtained using gamma (16%), generalized gamma (42%), and log-logistic (20%) functions provided the best fit to the EFS, OS, and TTNT data, respectively.

In both arms, the cure fraction was simultaneously estimated using logistic regression. More details about the extrapolation of the EFS, OS, and TTNT were provided in a previously published study [12].

### 2.6. Adverse Events

Due to their clinical relevance, only grade 3 or higher cytokine release syndrome (CRS) and neurotoxicity events observed in patients treated with axi-cel were considered in the analysis to evaluate the economic implications for their management. The incidence rates (6.5% CRS and 21.2% neurological events) were obtained from the ZUMA-7 clinical trial [11].

### 2.7. Utilities

The utility values associated with the event-free and post-event health states were considered to estimate QALYs (Table 1). For event-free patients, their quality of life was disaggregated into the periods before and after treatment with axi-cel or salvage immunochemotherapy. The utility values that were obtained from the literature were used in previous health technology assessments [12,22].

For patients who were alive and event-free for at least five years, the utility values of an age- and sex-matched Spanish general population were considered [23]. No disutility values due to CRS or neurological events were applied, as their potential influence on the quality of life was assumed to be captured by the on-treatment utility values.

### 2.8. Resource Use and Costs

In accordance with the NHS perspective, only direct health care costs were considered, which included drug acquisition and drug administration costs, disease management and monitoring costs, CRS and neurological event management costs, and end-of-life care costs (Table 2).

According to the clinical practice expressed by haematologists, 97.0% of patients who were candidates for treatment with axi-cel were assumed to undergo leukapheresis, 36.1% were assumed to receive bridging therapy (composed of salvage chemotherapy), 91.0% were assumed to receive conditioning chemotherapy, and 90.0% were assumed to receive CAR-T-cell infusion.

In the SoC arm, 100% of patients were treated with salvage immunochemotherapy (19% with R-DHAP, 42% with R-ESHAP, 31% with R-GDP, and 8% with R-ICE), 50% underwent stem cell harvest, 35.8% received high-dose chemotherapy, and 34.6% underwent ASCT, based on clinical practice.

Subsequent treatment costs in the model were applied as a one-off cost at the time of the initiation of third-line therapy based on the TTNT curve. Based on clinical practice in Spain, subsequent treatments after progression with axi-cel included chemotherapy (55%), inclusion in clinical trials (25%), and palliative treatment (20%). As CAR T-cell therapies are approved in the third-line setting, patients who progress after the SoC could receive axi-cel (35% of patients) or tisagenlecleucel (35% of patients). In addition, 10% of patients received palliative treatment after the SoC. An estimated average number of chemotherapy and palliative treatment cycles was derived for each subsequent therapy based on the observational evidence provided by haematologists; therefore, the duration of treatment was not explicitly modelled. Patients treated with axi-cel or the SoC who remained in an event-free state for five years were assumed to not require additional lymphoma treatment.

The LBCL management costs were estimated based on health care resource consumption, which depended on whether the patient had event-free or post-event disease and the treatment received (axi-cel or SoC). Disease management costs were mostly related to specialist visits, computerized tomography scans, and laboratory tests (Table 2).

The costs associated with adverse events ≥ grade 3 included treatment with tocilizumab and corticosteroids. The end-of-life care costs were applied as a one-off cost for each of the death events.

The drug acquisition costs were estimated based on published ex-factory prices [24], with a national mandatory deduction applied when necessary [25]. The unitary costs were obtained from a national health care cost database [26] and the literature [27]. Costs were expressed in euros, at 2022 values, and for those costs obtained from the literature, the cost was inflated to 2022 euros based on the Spanish general consumer price index [28].

### 2.9. Sensitivity Analysis

A deterministic one-way sensitivity analysis (OWSA) and a probabilistic sensitivity analysis (PSA) were performed to evaluate the robustness of the model and to determine the uncertainty surrounding the key model parameters. For the OWSA, the following parameters varied: the discount rate (0% and 5%); the standardized mortality ratio to reflect the higher rates of death in long-term responders (1.09) [29]; the proportion of patients treated with axi-cel (94.4%) according to ZUMA-7 [11]; the utility values from the ZUMA-7 study; the base case utility values (± standard deviation; ASCT cost (±20%); and the health care resource unitary costs (±20%).

The PSA was performed by the Monte Carlo method, and the model was run 5000 times. All uncertain parameters were simultaneously varied based upon appropriate probability distributions: beta distributions were applied to the utility values and proportions, log-normal distributions were applied to the adverse event frequency, and multivariate normal distributions were applied to correlated parameters, such as survival curves and gamma distributions for costs and resource use.

## 3. Results

### 3.1. Base Case

Over a lifetime horizon, axi-cel yielded 10.00 LYG per patient, which was greater than the LYG obtained with the SoC (8.28 LYG per patient), resulting in 1.72 additional LYG per patient treated with axi-cel vs. the SoC (Table 3).

In terms of quality of life, axi-cel provided more QALYs per patient (7.85 QALYs) than the SoC did (6.04 QALYs/patient), resulting in 1.81 additional QALYs per patient with axi-cel vs. the SoC (Table 3).

Over a lifetime horizon, patients treated with axi-cel incurred a total cost of EUR 343,581 per patient compared to the EUR 257,994 per patient treated with the SoC, representing an incremental cost of EUR 85,587. Axi-cel-related costs were driven primarily by the acquisition and administration costs of the CAR T-cell therapy, whereas the SoC-related costs were driven by subsequent treatments, including the CAR T-cell therapies administered in third-line treatments. The incremental subsequent treatment total costs of the SoC vs. axi-cel resulted in EUR 166,034 over the lifetime horizon.

The resulting incremental cost-effectiveness ratio of axi-cel vs. the SoC was 49,627 EUR/LYG, and the incremental cost-utility ratio (ICUR) was 47,309 EUR/QALY. Considering a willingness-to-pay threshold of up to 60,000 EUR per QALY [30], axi-cel could be a cost-effective alternative in the treatment of LBCL patients who are refractory or relapse after first-line therapy.

### 3.2. Sensitivity Analysis

In the OWSA, the inputs with the greatest influence on the results were the discount rate and the proportion of patients treated with axi-cel in the ZUMA-7 clinical trial. In 11 of 16 tested scenarios, the ICUR of axi-cel vs. the SoC was less than 50,000 EUR/QALY gained (Figure 2).

The PSA results were consistent with the base case results in terms of total costs and the QALYs gained. Axi-cel, compared with the SoC, was associated with a mean ICUR of 52,953 EUR/QALY (median of 46,740 EUR/QALY; IQR of 33,454–72,146 EUR/QALY) (Table 4). To show the PSA results, a cost-effectiveness plane (Figure 3) and a cost-effectiveness acceptability curve (Figure 4) were used.

## 4. Discussion

A cost-effectiveness analysis is an important tool that provides useful information for health care decision-makers on the adoption of new therapies.

The findings of this analysis have important implications for the adoption and reimbursement of axi-cel therapy in Spain. The results obtained in the base case analysis suggest that axi-cel could improve long-term health outcomes in the second-line treatment of LBCL patients who are refractory or have relapsed within 12 months of their first-line therapy and increase the LYG in pre-event with durable remission cases. Based on the ICUR obtained (47,309 EUR/QALY) and in the absence of an official willingness-to-pay threshold in Spain, axi-cel could be a cost-effective option [31] in the treatment of LBCL patients who are refractory or relapse after first-line therapy.

To our knowledge, this is the first study to evaluate the efficiency of axi-cel in comparison to salvage immunochemotherapy followed by high-dose chemotherapy and ASCT in the second-line treatment of patients with R/R LBCL in Spain. However, the efficiency of axi-cel vs. the SoC in the same indication (second-line) was evaluated under the United States payer’s perspective, using the same PS-MCM [12]. The study showed that axi-cel yielded, over a lifetime horizon, 1.51 incremental QALYs and USD 100,366 incremental total costs (costs reported in 2021 USD) compared with the SoC, resulting in an ICUR of 66,381 USD/QALY gained. Another study assessing the second-line treatment of R/R DLBCL conducted from the Swedish health care system perspective showed that axi-cel was cost-effective compared to the SoC, with an ICUR of 534,704 SEK/QALY (50,303 EUR/QALY) (costs reported in 2022 SEK) when assuming a willingness-to-pay threshold of 1,000,000 SEK/QALY (94,077 EUR/QALY) [32].

To reflect the cost-effectiveness analysis of Spanish clinical practice, the resource use, the unitary costs, and some of the patient characteristics were aligned with those of Spanish clinical experts who participated in the model validation process. Deterministic and probabilistic sensitivity analyses were conducted to analyse the effect of using alternative inputs and assumptions. Their results demonstrated the robustness of the model. Another strength of our analysis is that the model utilized data from the head-to-head randomized clinical trial ZUMA-7. It is important to highlight that further follow-up data were available for ZUMA-7 [33] at the time of writing this manuscript. At a median follow-up of 47.2 months, the median OS was not reached in the axi-cel group, and it was 31.1 months in the SoC arm, with estimated 4-year OS rates of 54.6% and 46.0%, respectively [33]. These long-term results confirmed the survival extrapolations made by our model, thus confirming the validity of the results obtained in the cost-effectiveness analysis.

Due to the nature of economic models, it is important to acknowledge the limitations of a cost-effectiveness analysis based on efficacy data from a clinical trial to ensure that the results are interpreted and applied appropriately in decision-making processes. The efficacy data obtained from the ZUMA-7 clinical trial might not accurately reflect the real-world effectiveness of axi-cel. This type of study often has strict inclusion and exclusion criteria, resulting in a study population that might not represent the broader Spanish patient population with LBCL. One of the main differences observed between clinical trials and clinical practice was the proportion of patients who received bridging therapy before their axi-cel infusion. According to haematologists, the proportion of patients with LBCL receiving bridging therapy in clinical practice is likely to double that observed in the ZUMA-7 study (36.1%). In addition, in the analysis, bridging therapy was composed of a pool of salvage chemotherapy; however, in the ZUMA-7 study, it was composed only of corticosteroids.

The current cost-effectiveness analysis revealed that the utility value data obtained in the literature and used to represent quality of life were applicable to the Spanish health care setting without any adaptation, which could represent a limitation. This assumption was validated by haematologists and considered appropriate. Another key parameter was subsequent third-line treatment, for which the proportion of CAR T-cells considered in the model was determined according to clinical practice (70%). This proportion differs from that observed in the ZUMA-7 clinical trial (56%); however, it is possible that both proportions might not reflect the treatment patterns in some autonomous communities of Spain. Finally, the present analysis might not capture all the relevant factors that could influence the cost-effectiveness analysis of axi-cel in real-world practice. The differences between Spanish health care areas, resource availability, the adverse events associated with each treatment, and patient preferences could affect the generalizability of the results obtained.

Despite the limitations detailed, the results obtained in the OWSA and the PSA confirmed the robustness of the model. In the OWSA, the results obtained when the key parameters and assumptions were tested did not show a great variation with respect to the base case ICUR. The tornado diagram showed that the most influential parameters could be the discount rate and the proportion of patients who finally received an axi-cel infusion (after the leukapheresis, the bridging therapy, and the conditioning chemotherapy), which are intrinsically related to its acquisition cost, estimated based on the list price.

The results obtained in the present analysis highlight the effectiveness of axi-cel compared to the SoC for LBCL treatment after first-line chemotherapy. Although the initial costs related to axi-cel are higher than those related to the SoC, health outcomes are achieved over a lifetime horizon, and subsequent treatment costs are avoided, confirming its use in patients who have not responded to first-line therapy. The inclusion of a cost-effectiveness analysis in the reimbursement process would help to promote the efficiency and financial sustainability of the Spanish NHS. The findings obtained in the present study are an important tool that provides useful information for health care decision-makers to find ways to increase the affordability and accessibility of CAR T-cell therapy. Axi-cel has demonstrated that it could be a cost-effective option, and these results would allow clinicians and other stakeholders to make informed decisions regarding the most appropriate treatment for LBCL. 

Further research and real-world patient-level data collection on the use of axi-cel in the second-line setting could refine the analysis, reduce the uncertainties, and strengthen the evidence base on the cost-effectiveness of axi-cel therapy in R/R LBCL. In addition, it would be interesting to include in this type of health economic evaluation the ability of this information to estimate the expected return on investment for the finite funding of comparable alternatives.

## 5. Conclusions

This cost-effectiveness analysis demonstrated that, compared with the SoC, axi-cel could improve health outcomes in terms of LYG and QALYs in the second-line treatment of LBCL patients who were refractory or relapsed within 12 months of first-line therapy.

## Figures and Tables

**Figure 1 cancers-16-02301-f001:**
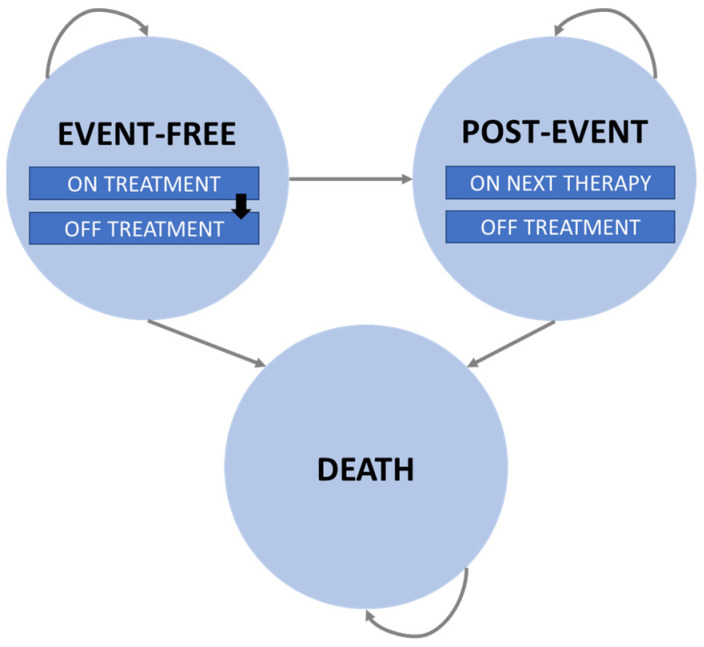
Decision-analytic model structure.

**Figure 2 cancers-16-02301-f002:**
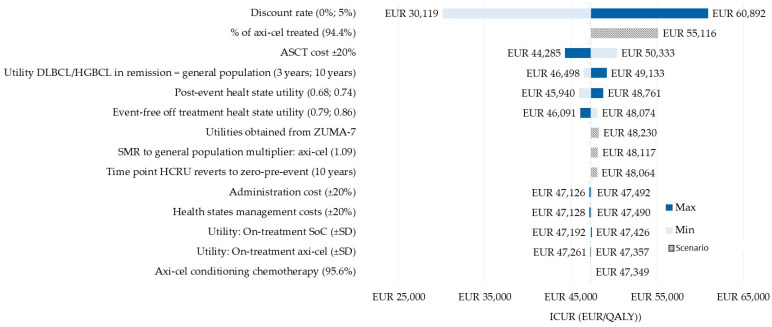
Tornado diagram showing the most influential parameters. ASCT, autologous stem cell transplantation; axi-cel, axicabtagene ciloleucel; DLBCL, diffuse large B-cell lymphoma; EFS, event-free survival; HGBCL, high-grade B-cell lymphoma; ICUR, incremental cost-utility ratio; QALY, quality-adjusted life year.

**Figure 3 cancers-16-02301-f003:**
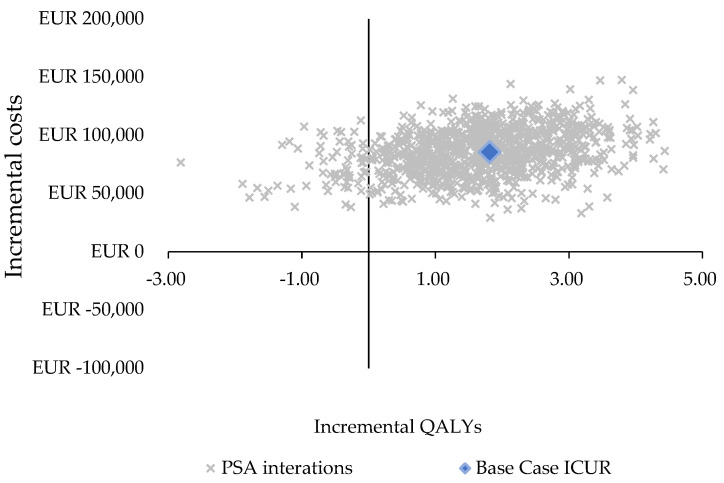
Cost-effectiveness plane. ICUR, incremental cost-utility ratio; PSA, probabilistic sensitivity analysis; QALY, quality-adjusted life year.

**Figure 4 cancers-16-02301-f004:**
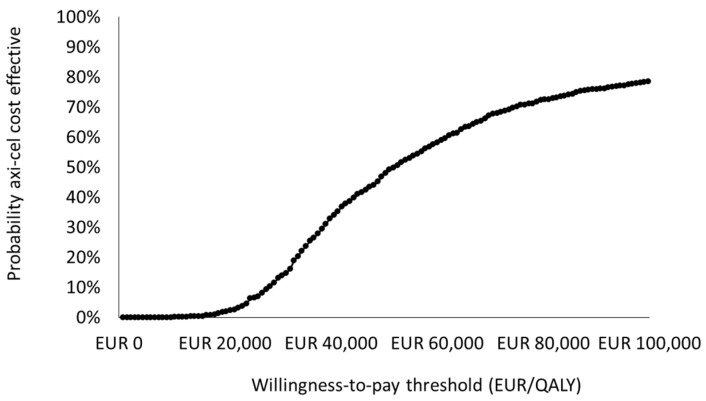
Cost-effectiveness acceptability curve. ICUR, incremental cost-utility ratio; PSA, probabilistic sensitivity analysis; QALY, quality-adjusted life year.

**Table 1 cancers-16-02301-t001:** Utility values.

Health States	Utility Value	Reference
Event-free: on treatment with axi-cel (one cycle)	0.74	Roth et al., [22]
Event-free: on treatment with SoC (three cycles)	0.67	Roth et al., [22]
Event-free: off treatment	0.82	Roth et al., [22]
Post-event	0.71	TA567 [23]

**Table 2 cancers-16-02301-t002:** Drug and health care resource costs (EUR, 2022).

Axi-Cel-Related Costs	
Acquisition cost	EUR 313,920 *
Leukapheresis	EUR 1025
Bridging therapy	EUR 2599
Conditioning chemotherapy	EUR 1249
Administration and monitoring	EUR 9794
SoC-related costs	
Salvage immunochemotherapy	EUR 4063
Administration	EUR 2443
Stem cell harvest	EUR 1025
High-dose chemotherapy	EUR 9205
ASCT (procedure and annual monitoring)	EUR 79,358
Subsequent treatment total cost	
After axi-cel	EUR 32,754
After SoC	EUR 233,412
Health states management costs	
Event-free with axi-cel (EUR/month)	EUR 305
Event-free with SoC (EUR/month)	EUR 527
Post-event with axi-cel (EUR/month)	EUR 537
Post-event with SoC (EUR/month)	EUR 352
Adverse event grade ≥ 3 management costs	
Cytokine release syndrome (per-event cost)	EUR 2077
Neurological events (per-event cost)	EUR 24
End-of-life care costs (one-off cost)	EUR 6267

* Axi-cel cost was estimated taking into account the list price and the mandatory deduction of 4%. ASCT, autologous stem-cell transplantation; axi-cel, axicabtagene ciloleucel; SoC, standard of care.

**Table 3 cancers-16-02301-t003:** Base case results.

	Axi-cel	SoC	Incremental (axi-cel vs. SoC)
Total LYG	10.00	8.28	1.72
LYG in event-free state	7.10	3.12	3.98
LYG in post-event state	2.90	5.15	−2.25
Total QALY	7.85	6.04	1.81
QALY in event-free state	5.90	2.57	3.33
QALY in post-event state	1.95	3.47	−1.52
Total costs per patient	EUR 343,581	EUR 257,994	EUR 85,587
Axi-cel-related costs	EUR 294,326	EUR 0	EUR 294,326
SoC-related costs	EUR 0	EUR 40,889	EUR –40,889
Subsequent treatment	EUR 18,598	EUR 184,632	EUR –166,034
Health state management	EUR 26,112	EUR 27,748	EUR −1636
AEs management	EUR 140	EUR 0	EUR 140
Palliative care	EUR 4406	EUR 4726	EUR −319
ICER (axi-cel vs. SoC)		49,626 EUR/LYG	
ICUR (axi-cel vs. SoC)		47,308 EUR/QALY	

Axi-cel, axicabtagene ciloleucel; ICER, incremental cost-effectiveness ratio; ICUR, incremental cost-utility ratio; LYG, life years gained; QALY, quality-adjusted life year; SoC, standard of care.

**Table 4 cancers-16-02301-t004:** Probabilistic sensitivity analysis results.

	Axi-cel vs. SoC
Incremental QALY	1.61
Incremental costs	EUR 85,433
ICUR (mean)	52,953 EUR/QALY
ICUR (median, IQR)	46,740 EUR/QALY (33,454–72,146 EUR/QALY)

Axi-cel, axicabtagene ciloleucel; ICUR, incremental cost-utility ratio; IQR, interquartile range; QALY, quality-adjusted life year; SoC, standard of care.

## Data Availability

The data presented in this study are available in this article.

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
