# Peer review of "Cost-Effectiveness Analysis of Axicabtagene Ciloleucel vs. Standard of Care in Second-Line Treatment for Relapsed/Refractory Large B-Cell Lymphoma in Spain"

_cancers, 2024, doi:10.3390/cancers16132301_

Round 1

Reviewer 1 Report

Comments and Suggestions for Authors

1. How has the treatment landscape for relapsed or refractory large B-cell lymphoma (R/R LBCL) evolved in recent years, and what prompted the study to assess the cost-effectiveness of axi-cel compared to the standard of care in Spain?

2. Considering the promising results of axi-cel treatment for R/R DLBCL, is it a cost-effective alternative to the current standard of care (SoC) involving salvage immunochemotherapy and ASCT in the Spanish healthcare system?

3. While axi-cel demonstrates improved event-free survival (EFS) and response rates compared to SoC, does the study explore the potential for cure with each treatment option?

4. The analysis uses data with a median follow-up of 24.9 months. Does the study address the long-term cost-effectiveness of axi-cel compared to SoC given the potential for future treatment needs?

5. Beyond cost and effectiveness, are there other factors, such as treatment side effects or patient preferences, that should be considered when choosing between axi-cel and SoC in Spain?

6. What are the main cost and health outcome differences observed between patients treated with axi-cel and those treated with HDT+ASCT for relapsed/refractory large B-cell lymphoma, and what does the incremental cost-utility ratio indicate about the cost-effectiveness of axi-cel in Spain?

7. The abstract mentions a "partitioned survival mixture-cure model" to estimate the cost-effectiveness. Can this model account for the potential for cure with each treatment option?

Comments on the Quality of English Language

It sems fine for publication.

Author Response

  1. How has the treatment landscape for relapsed or refractory large B-cell lymphoma (R/R LBCL) evolved in recent years, and what prompted the study to assess the cost-effectiveness of axi-cel compared to the standard of care in Spain?

In line with the reviewer comment, we have added new information about the landscape for R/R LBCL patients before the axi-cel availability:

“Despite the significant advances in the therapeutic management of B-cell lympho-mas, approximately 30%-40% of patients with DLBCL do not respond or relapse after standard first-line treatment with R-CHOP (rituximab in combination with cyclophos-phamide, doxorubicin, vincristine, and prednisone). Until recently, the standard of care (SoC) for second-line treatment with curative in-tent for patients with relapsed or refractory (R/R) LBCL was salvage immunochemother-apy followed by high-dose chemotherapy and autologous stem-cell transplantation (ASCT), in chemosensitive patients. Unfortunately, only 50% of patients respond to second-line salvage immunochemotherapy and can undergo ASCT, with a 3-year event-free survival (EFS) rate of 31%.”

Regarding the second question, the development of the study was mainly motivated by the significant clinical improvements observed in clinical trial and because economic evaluations are an important tool that provide useful information for health decision-makers in the adoption of these type of innovative therapies. We have modified the objective of our study to reflect the reviewer's suggestion:

“Given the significant improvements observed with axi-cel and economic evaluations have become an important tool for decision makers, this study aimed to assess the cost-effectiveness of axi-cel compared with SoC as a second-line treatment for R/R LBCL in Spain.”

  1. Considering the promising results of axi-cel treatment for R/R DLBCL, is it a cost-effective alternative to the current standard of care (SoC) involving salvage immunochemotherapy and ASCT in the Spanish healthcare system?

Currently there is not an official willingness-to-pay threshold in Spain. The available data suggest that it could be reasonable to use thresholds of 25,000 € and 60,000 € per QALY, therefore axi-cel could be a cost-effective option.

This information is explained in discussion epigraph: “Based on the ICUR obtained (€47,309/QALY) and in the absence of an official willingness-to-pay threshold in Spain, axi-cel could be a cost-effective option in the treatment of LBCL patients who are refractory or relapse to first-line therapy”.

  1. While axi-cel demonstrates improved event-free survival (EFS) and response rates compared to SoC, does the study explore the potential for cure with each treatment option?

Regarding the reviewer question, the partitioned survival mixture cure model (the model exployed to developed the cost-effectiveness analysis) was the most accurate approach for predicting OS over the long term, as it allows presenting a cohort composed of a proportion of patients who were “statistically cured” with an OS similar to that of the age- and sex-matched general population and a proportion of patients “not cured” who were at increased risk of mortality related to R/R LBCL. For axi-cel and SoC, the cure fraction was estimated using logistic regression. So, definitely, yes, de study explores the potential for cure with each treatment option.

These arguments were explained in epigraph 2.1 Model Structure, 2.4 Clinical data and 2.5 Statistical methods.

  1. The analysis uses data with a median follow-up of 24.9 months. Does the study address the long-term cost-effectiveness of axi-cel compared to SoC given the potential for future treatment needs?

Although the model uses the EFS and OS data from the ZUMA-7 clinical trial with a median follow-up of 24.9 months, the curves were extrapolated beyond this period to estimate the EFS and OS data over a lifetime horizon. Currently, new data for ZUMA-7 are available and, at a median follow-up of 47.2 months, the median OS was not reached in the axi-cel group and was 31.1 months in the SoC arm, with estimated 4-year OS rates of 54.6% and 46.0%, respectively. These long-term results are in line with the survival extrapolations made in our model, thus maintaining the validity of the results obtained in the cost-effectiveness analysis.

In addition, the analysis has considered the subsequent treatment administered after possible progression with axi-cel or with salvage immunochemotherapy + ASCT, so that future treatments to be administered were also included.

  1. Beyond cost and effectiveness, are there other factors, such as treatment side effects or patient preferences, that should be considered when choosing between axi-cel and SoC in Spain?

We appreciated the reviewer comments which have made us reflect on the choice of treatment. New information has added to the discussion:

“Despite the health benefits and the efficiency demonstrated by axi-cel compared with SoC, other factors should be considered, such as the possible adverse events associated with each treatment or the patient´s preferences, which could be influenced by various conditions.”

  1. What are the main cost and health outcome differences observed between patients treated with axi-cel and those treated with HDT+ASCT for relapsed/refractory large B-cell lymphoma, and what does the incremental cost-utility ratio indicate about the cost-effectiveness of axi-cel in Spain?

As we mention in the results, the main differences in costs between axi-cel and SoC was derived by the second-line treatment costs and the subsequent treatment administered after progression. To clarify this point, we have modified the wording in the description of results:

“Axi-cel-related costs were driven primarily by acquisition and administration costs of CAR T therapy, whereas SoC-related costs were driven by subsequent treatment, including CAR T-cell therapies administered in third-line.

Regarding the health outcomes, axi-cel provided 1.81 additional QALYs per patient versus SoC. Considering the incremental costs and the incremental QALYs, axi-cel could be a cost-effectiveness alternative compared with SoC, with an incremental cost-utility ratio of €47,309/QALY. We have added new information about the threshold in the results:

“The resulting incremental cost-effectiveness ratio of axi-cel versus SoC was €49,627/LYG, and the incremental cost-utility ratio (ICUR) was €47,309/QALY. Considering a willingness-to-pay thresholds up to €60,000 per QALY, axi-cel could be a cost-effectiveness alternative in the treatment of LBCL patients who are refractory or re-lapse to first-line therapy”.

  1. The abstract mentions a "partitioned survival mixture-cure model" to estimate the cost-effectiveness. Can this model account for the potential for cure with each treatment option?

This type of model has been widely used in the oncology area, and specifically to represent the survival of patients treated with CAR T therapies. Mixture cure models allow estimating the probability of survival of a cohort which is composed of “cured” and “uncured” patients, in order to provide accurate survival estimates in the presence of statistical cure.

We have added to the text the reference on why to use this type of models in oncology (Felizzi F, Paracha N, Pöhlmann J, et al. Mixture Cure Models in Oncology: A Tutorial and Practical Guidance. Pharmacoecon Open. 2021;5(2):143-155.)

Reviewer 2 Report

Comments and Suggestions for Authors

This paper presents a cost-effectiveness analysis of axicabtagene ciloleucel (axi-cel) versus standard of care (SoC) in the second-line treatment of relapsed/refractory large B-cell lymphoma (R/R LBCL) in Spain. Utilizing a partitioned survival mixture-cure model, the study estimates the costs, life-years gained (LYG), and quality-adjusted life years (QALY) over a lifetime horizon. The findings suggest that axi-cel offers superior health outcomes compared to SoC, with incremental cost-effectiveness and cost-utility ratios indicating that axi-cel could be a cost-effective option in the Spanish healthcare setting. A few questions:

1. The sensitivity analysis shows the robustness of the model. Could the authors discuss any scenarios where axi-cel might not be considered cost-effective? What were the key drivers of cost-effectiveness in the sensitivity analyses?

2. The PSA was run 5,000 times. Could the authors provide more details on the distribution of the results? Were there any notable patterns or outliers that emerged from this analysis?

3. The cost analysis includes subsequent treatments post-axi-cel and SoC. How did the cost and frequency of these treatments compare between the two groups?

4. What are the policy implications of these findings for the Spanish National Health System (NHS)? How might this study influence decisions on funding and accessibility of CAR T-cell therapies?

Comments on the Quality of English Language

Overall looks good.

Author Response

This paper presents a cost-effectiveness analysis of axicabtagene ciloleucel (axi-cel) versus standard of care (SoC) in the second-line treatment of relapsed/refractory large B-cell lymphoma (R/R LBCL) in Spain. Utilizing a partitioned survival mixture-cure model, the study estimates the costs, life-years gained (LYG), and quality-adjusted life years (QALY) over a lifetime horizon. The findings suggest that axi-cel offers superior health outcomes compared to SoC, with incremental cost-effectiveness and cost-utility ratios indicating that axi-cel could be a cost-effective option in the Spanish healthcare setting. A few questions:

  1. The sensitivity analysis shows the robustness of the model. Could the authors discuss any scenarios where axi-cel might not be considered cost-effective? What were the key drivers of cost-effectiveness in the sensitivity analyses?

In line with the reviewer questions, we have discussed the key drivers of cost-effectiveness analysis and the results obtained in the OWSA. Additional information was added to discussion:

“In OWSA, the results obtained when the key parameters and assumptions were tested did not show a great variation with respect to the base case ICUR. The tornado diagram showed that the most influential parameters could be the discount rate and the proportion of patients who finally received the axi-cel infusion (after the leukapheresis, the bridging therapy and the conditioning chemotherapy), which is intrinsically related to its acquisition cost estimated based on the list price.”

  1. The PSA was run 5,000 times. Could the authors provide more details on the distribution of the results? Were there any notable patterns or outliers that emerged from this analysis?

In line with this comment and as suggested by the other reviewers, more information on PSA results was added to the manuscript. We considered appropriate to present the results of the PSA as mean, median, and IQR (P25 - P75).

“The PSA results were consistent with the base case results in terms of total costs and QALYs gained (Figure 3). Axi-cel compared with SoC was associated with a mean ICUR of €52,953/QALY (median €46,740/QALY; IQR €33,454/QALY - €72,146/QALY) (Table 4).”

Furthermore, we added the cost-effectiveness acceptability curve (Figure 4) and a new table (Table 4) with the PSA results. After 5,000 simulations, the mean (€52,953/QALY) and median (€46,740/QALY) ICUR was similar to the deterministic result (€47,309/QALY), representing the absence of heterogeneity in the parameters included in the model. Furthermore, the representation of each PSA iteration in the cost-effectiveness plane as a “cloud” allows to observe the low variability with respect to the base case results and the low dispersion between iterations.

  1. The cost analysis includes subsequent treatments post-axi-cel and SoC. How did the cost and frequency of these treatments compare between the two groups?

As we detailed in the Resource Use and Costs section (from line 227 onwards) subsequent treatment costs in the model were applied as a one-off cost at the time of initiation of third-line therapy based on the TTNT curve. An estimated average number of treatment cycles (in case of chemotherapy) was derived for each subsequent therapy based on observational evidence; thus, the duration of treatment was not explicitly modeled. As CAR T-cell therapies are approved in the third-line setting, patients who progress after SoC could receive axi-cel or tisagenlecleucel.

The results showed that SoC-related costs were driven by subsequent treatment, including CAR T-cell therapies administered in third-line. The incremental subsequent costs of SoC versus axi-cel was €166,034 over a lifetime horizon. Therefore the differences in subsequent treatment mix between the two treatment arms were also a key driver of the results.

In line with the reviewer comment, additional information about the subsequent treatment was added to “Resource Use and Costs” and “Results” section.

  1. What are the policy implications of these findings for the Spanish National Health System (NHS)? How might this study influence decisions on funding and accessibility of CAR T-cell therapies?

We appreciate the reviewer comment that made us reflect on the implications of the results obtained. We have added the following sentence to the discussion section:

“The inclusion of cost-effectiveness analysis in the reimbursement process would help to promote efficiency and financial sustainability of the Spanish NHS. The findings obtained in the present study are an important tool that provides useful information for health care decision-makers to find ways to increase the affordability and accessibility of CAR T-cell therapy. Axi-cel has demonstrated that could be a cost-effective option and these results would allow clinicians and other stakeholders to make informed decisions regarding the most appropriate treatment of LBCL.”

Reviewer 3 Report

Comments and Suggestions for Authors

The manuscript describes model-based cost/effectiveness analysis of axicabtagene ciloleucel versus standard of care in second-line treatment for relapsed/refractory large B-cell lymphoma in Spain. The manuscript has sufficient novelty, and there is practical significance of the results. Majority of the study was methodologically correct; however, there are some issues that the authors should address to improve the analysis and the manuscript:

1. protocol of the study should have been preregistered and made publicly available; since this was not the case, the author should offer some explanation:

2. the first-order uncertainty was not taken into account when presenting results of the output parameters in base case; please add the confidence intervals to the output parameters.

3. Values of the output parameters after PSA should be presented in a table, so a reader could understand the differences from the base case.

4. Value of information analysis should be added to the manuscript, showing values of the Expected Value of Perfect information (EVPI), the Expected Value of Perfect Parameter Information (EVPPI), Expected Value of Sample Information (EVSI) and Expected Net Gain of Sampling (ENGS).

5. A paragraph about Limitations of the study should be added to the Discussion section.

6. The Acceptability curve should be presented

7. An explanation should be given why differential sampling of costs and QALYs was not used, and why 3% discount rate was chosen

Author Response

The manuscript describes model-based cost/effectiveness analysis of axicabtagene ciloleucel versus standard of care in second-line treatment for relapsed/refractory large B-cell lymphoma in Spain. The manuscript has sufficient novelty, and there is practical significance of the results. Majority of the study was methodologically correct; however, there are some issues that the authors should address to improve the analysis and the manuscript:

  1. protocol of the study should have been preregistered and made publicly available; since this was not the case, the author should offer some explanation:

In line with the reviewer suggestions, we would like to highlight that the development of the study was fully described in the article. We have modified the next sentence:

“Although there is no specific protocol for this study, more details about the model development were previously published [12].”

  1. the first-order uncertainty was not taken into account when presenting results of the output parameters in base case; please add the confidence intervals to the output parameters.

We appreciate the reviewer comment, however there is no variability in the deterministic results (base case results).  On the other hand, in the cost-effectiveness plane (graphic representation of the results obtained in the PSA) we can observed that the results of each of the 5,000 simulations do not follow a normal distribution. Because the standard deviation is a measure that is very sensitive to extreme values, we considered appropriate to present the results of the PSA as mean, median, and IQR (P25 - P75). To provide this information, we have added new sentences in the sensitivity analysis results section:

“The PSA results were consistent with the base case results in terms of total costs and QALYs gained (Figure 3). Axi-cel compared with SoC was associated with a mean ICUR of €52,953/QALY (median €46,740/QALY; IQR €33,454/QALY - €72,146/QALY) (Table 4).”

  1. Values of the output parameters after PSA should be presented in a table, so a reader could understand the differences from the base case.

According with the reviewer suggestion, we have added to the manuscript the table 4 with the PSA results.

Table 4. Probabilistic sensitivity analysis results

Axi-cel vs. SoC

Incremental QALY

1.61

Incremental costs

85,433

ICUR (mean)

€52,953/QALY

ICUR (median, IQR)

€46,740/QALY (€33,454/QALY – €72,146/QALY)

  1. Value of information analysis should be added to the manuscript, showing values of the Expected Value of Perfect information (EVPI), the Expected Value of Perfect Parameter Information (EVPPI), Expected Value of Sample Information (EVSI) and Expected Net Gain of Sampling (ENGS).

We appreciated the reviewer comments however, although the inclusion of value of information as a part of health economics evaluations is increasing, our purpose was assessing the effectiveness of the axi-cel vs. SoC estimating the incremental cost-utility ratio. It would be very interesting for future projects to analyze the value of information to estimate the greatest expected return for finite funding.

A sentence about the value of information was added to discussion:

“Further research and real-world data collection on the use of axi-cel in the second-line setting could refine the analysis, reduce the uncertainties, and strengthen the evidence base on the cost-effectiveness of axi-cel therapy in R/R LBCL. In addition, it would be in-teresting to include in this type of health economic evaluations, the value of information to estimate the expected return on investment for finite funding of comparable alternatives.”

  1. A paragraph about Limitations of the study should be added to the Discussion section.

In line with the reviewer comment, we have modified some sentences in the discussion to reflect the limitations of our analysis. Limitations are extensively detailed in two long paragraphs, from line 360 to 384.

  1. The Acceptability curve should be presented.

According with the reviewer suggestion, the cost-effectiveness acceptability curve was presented (Figure 4).

  1. An explanation should be given why differential sampling of costs and QALYs was not used, and why 3% discount rate was chosen.

We appreciate the reviewer suggestion, however we do not understand what the reviewer means by “differential sampling”. If by this type of study, the reviewer means to select different subgroups based on certain characteristics to explore variations in economic and health outcomes, we would like to point out that a study of this type was not possible because it would require the availability of patient-level data, and in our case was not possible.

Regarding the discount rate, as we detailed in the 2.1 epigraph, the discount rate of 3% was considered taking into account the recommendations for conducting economic evaluations in Spain [López Bastida J, Oliva J, Antoñanzas F, et al. Propuesta de guía para la evaluación económica aplicada a las tecnologías sa-nitarias [Guide proposal for economic evaluation applied to health technologies]. Gac Sanit. 2010;24(2):154–170. Spain].

Reviewer 4 Report

Comments and Suggestions for Authors

Dear Authors,

Thank you for submitting this article! It reveals important information for Spanish National Health System.

I would suggest to move some paragraphs from 2.4 Clinical data to new section named Statistical methods.

Author Response

Dear Authors,

Thank you for submitting this article! It reveals important information for Spanish National Health System.

I would suggest to move some paragraphs from 2.4 Clinical data to new section named Statistical methods.

According with the reviewer suggestion, the information about the statistical methods employed to adjust and extrapolate the clinical data was moved to a new epigraph named “2.5 Statistical methods” and situated under the epigraph “2.4 Clinical data”.